# Detecting and Understanding Self-Deleting JavaScript Code

## ABSTRACT

Self-deletion is a well-known strategy frequently utilized by malware to evade detection. Recently, this technique has found its way into client-side JavaScript code, significantly raising the complexity of JavaScript analysis. In this work, we systematically study the emerging client-side JavaScript self-deletion behavior on the web. We tackle various technical challenges associated with JavaScript dynamic analysis and introduce JSRay, a browser-based JavaScript runtime monitoring system designed to comprehensively study client-side script deletion. We conduct a large-scale measurement of one million popular websites, revealing that script self-deletion is prevalent in the real world. While our findings indicate that most developers employ self-deletion for legitimate purposes, we also discover that self-deletion has already been employed together with other anti-analysis techniques for cloaking suspicious operations in client-side JavaScript.

## CCS CONCEPTS

• **Security and privacy** → **Web application security**.

## KEYWORDS

JavaScript, anti-analysis techniques, web browser

**ACM Reference Format:**
Anonymous Author(s). 2024. Detecting and Understanding Self-Deleting JavaScript Code. In *Proceedings of the ACM Web Conference 2024 (WWW '24).* ACM, New York, NY, USA, 11 pages.

## 1 INTRODUCTION

As a traditional defense evasion and anti-analysis technique, self-deletion has been widely adopted in real-world malware samples [5, 7, 14]. In the cases of a binary malware, it could delete its executable file(s) from the system after launch at a victim host system, and keep residing in the system's memory to perform malicious operations. This technique can be used to bypass some host forensic analysis and anti-virus detections that scan the file systems and analyze the original executables. It has been observed in the past that multiple malware families (*e.g.*, Anchor [5] and Win32/Nemim.gen!A [14]) had adopted such anti-analysis technique to hide their malicious activities.

The self-deletion technique is also emerging in web applications, particularly in client-side JavaScript code. On the web, the client-side JavaScript is embedded in the page's HTML code or loaded from other URLs, and parsed by and executed in the browser's JavaScript engine. Once a JavaScript script starts execution in the JavaScript engine (just like a binary is loaded into the memory), its execution becomes independent from the original source code contained in the web page.

The basic anti-debugging technique (BADT) has been systematized by prior work [26]. As a new and under-studied anti-debugging technique, self-deletion is more effective and stealthier than existing methods. The use of self-deletion in client-side JavaScript can make it more difficult to study and analyze the dynamic behavior of the numerous scripts [18], that are widely included in today's websites. Nevertheless, it has been demonstrated that state-of-the-art malware detection or analysis tools cannot sufficiently detect and monitor such behaviors. Even if their behaviors are identified, the security analysts still need great manual work to locate the threats on the page once self-deletion is adopted.

In this work, we aim to systematically study the emerging client-side JavaScript self-deletion behavior on the web. We want to study the commonly used JavaScript self-deletion techniques and help people understand the security implication of self-deleting JavaScript code. In particular, we are interested in understanding how common this technique is applied in the real world and why some web developers use this technique in their code. Furthermore, since self-deletion can naturally be applied to cloak some suspicious or even malicious activities, we also try to investigate the connection between the uses of self-deletion and suspicious operations in real-world JavaScript code.

We face several technical challenges in detecting and analyzing self-deleting client-side JavaScript scripts. First, JavaScript code can be dynamically included and even generated at runtime, making it challenging to cover all the executed scripts. Second, as the deletion of a script removes its container in the Document Object Model (DOM) tree, we need to identify and map its independent JavaScript object in the JavaScript engine. Finally, self-deletion can be used along with other anti-analysis techniques, such as code obfuscation, to further cloak a script's operations.

We find the solutions of prior works [3, 15, 21] cannot satisfy our requirements and are incomplete in some aspects. Inspired by their methods, we develop a new browser-based JavaScript runtime monitoring system—JSRay—for comprehensively studying client-side script deletion. First, JSRay incorporates a runtime script inclusion monitor mechanism to dynamically monitor the inclusion and execution of all JavaScript code on a page. This inclusion monitoring mechanism can help track the origin of an included script and also construct the inclusion dependency among all the scripts. Second, JSRay also includes a runtime deletion monitoring mechanism to detect all the dynamic script deletion operations. It can further tag the scripts running in the JavaScript world to their corresponding containers in the page to accurately attribute the deletions. Third, JSRay also dynamically monitors JavaScript's access to sensitive browser APIs for helping study the connection between the self-deleting scripts and any suspicious behaviors. We implemented a

prototype of JSRᴀʏ on top of the Chromium browser and anonymously release its source code [2].

We applied the prototype of JSRᴀʏ in a large-scale measurement of one-million popular websites to study the prevalence of script deletion behaviors in the real world. Our measurement reveals that script deletion is quite common in the real world—about 42.44% of the websites in our dataset contained at least one dynamically deleted script. These websites were almost uniformly ranked, *i.e.*, we did not observe that the more popular websites included more/less deleted scripts. We further discover that more than half (61.39%) of the deletions were labeled as self-deletion—the deleted script was either removed directly by itself, or removed by a script that either loaded it or was loaded by it. This indicates that the developers might partition their code into separate scripts and dynamically clean some of them.

To understand what is deleted from the page, we performed a manual analysis of 600 randomly sampled self-deleting scripts. We found that 92% of those scripts were benign or legitimate scripts and the others were involved with suspicious activities, such as accessing and stealing user credentials stored in client-side storages, tracking user activities without user consent, and malvertising. To answer whether self-deletion has been widely used for hiding some unwanted, suspicious or even malicious behaviors, we leverage a popular filter list and our sensitive API monitor to analyze self-deleting and normal scripts. We discovered that the self-deleting scripts were 5.6x more likely to be blocked by the list and performed 11.8x more accesses to the sensitive browser APIs than the normal scripts, indicating that the self-deleting scripts are more suspicious.

In summary, our paper makes the following contributions.

- We developed a browser-based JavaScript runtime monitoring system—JSRᴀʏ, that can comprehensively monitor dynamic JavaScript operations.
- We are the first to systematically defined and studied self-deletion in client-side JavaScript on the web.
- We conducted a large-scale study on one million websites. Our results showed that self-deleting scripts are prevalent and they could be related to suspicious operations that threaten the end users.

## 2 PROBLEM STATEMENT

In this section, we describe our research scope and discuss our research challenges.

### 2.1 Research Scope and Assumptions

Self-deletion means that a program deletes its binary (file) or any other resources after it starts the execution. It can be used to bypass host forensic analysis and has already been adopted by real-world malware [5, 7, 14]. Self-deletion can also be used in client-side JavaScript: a script can delete its host HTML `<script>` element or any other host container (*e.g.*, an HTML attribute) from the Document Object Model (DOM) tree. Deleting the code container only detaches the script source from the DOM tree, and the *deleted* script still exists and runs in the JavaScript world.

In this work, we aim to study the use of self-deletion in JavaScript code as an anti-analysis technique and the corresponding security implication. We mainly study the client-side JavaScript and exclude the JavaScript in browser extensions and non-browser applications from our scope. We do not attempt to cover all the scripts that can be executed only under specific conditions or events, which has been known to be an important research problem in software testing [12, 18]. Instead, we target the automatically executed scripts for studying the self-deletion behaviors. This is reasonable as most JavaScript code (inline or external) is introduced by the HTML `<script>` tags, which are executed immediately when they are parsed by the browser.

From the security perspective, we attempt to discover the connection between the uses of the self-deletion technique and suspicious/malicious activities. Since client-side JavaScript's privilege is strictly restricted by modern browser security mechanisms (*e.g.*, sandbox and the SOP), traditional web attacks such as corrupting the browser memory for control-flow hijacking are rendered almost infeasible. Today's malicious scripts primarily threaten users' privacy through exfiltrating sensitive client-side user data (*e.g.*, username and password [31]) and tracking users' online activities [20, 33]. Therefore, we consider mainly malicious scripts that access sensitive client-side data. It is notable that detecting malicious JavaScript code, which has been extensively studied by prior work [8–11, 24, 37], is orthogonal to our research task. We do not attempt to detect whether the self-deleting scripts are malicious JavaScript. Rather, we aim to understand the client-side JavaScript self-deletion technique and figure out whether the use of it is suspicious.

### 2.2 Research Challenges

We face the following challenges in detecting and studying self-deleting scripts.

- **Dynamic Code.** As a dynamic programming language, client-side JavaScript code can be dynamically loaded or generated at runtime. It is challenging to monitor and cover all the executed JavaScript code statically [4]. Prior work [21] captures script inclusion relationship via only network requests. But some dynamically loaded scripts are inline scripts that do not have a source URL. Therefore, a new dynamic runtime monitoring approach would be needed.

- **Script Identification.** JavaScript code can be included and hence deleted via multiple ways in a web page. In order to detect the dynamically deleted scripts, we will need to study the different deletion techniques and monitor the relevant JavaScript operations to identify the deleted scripts. Unfortunately, prior works [3, 15] cannot completely track the origins and relationships of all kinds of scripts. Also, how to map a JavaScript object (asynchronously) running in the JavaScript engine to its DOM container would be difficult, because the deleting script can delete the container of another script instead of its own. Therefore, we need to distinguish the script performing deletion from the one being deleted.

- **Code Obfuscation.** Self-deleting scripts could be obfuscated in order to hide their (suspicious or malicious) behaviors. Obfuscation makes it difficult to identify the deletion operations and understand the other behaviors of the scripts. We need to find a robust approach to accurately monitor the dynamic behaviors of the obfuscated JavaScript code.

# 3  SCRIPT DELETION

In this section, we define different classes of script deletion operations and present the techniques for deleting a script in client-side JavaScript. Then we discuss the security impact of the script deletion techniques.

## 3.1  Classes of Script Deletion

Since all JavaScript code in one frame run with the same privilege in the browser, a script can be deleted by any other script in the same frame. Therefore, it is necessary to identify both the *deleting script* and the *deleted script*. We will discuss in §4 our method for identifying the deleting script and the deleted script in the browser.

Depending on the relationship of the two scripts, we can categorize script deletion into four classes: 1) deletion by itself—a script is deleted by itself; 2) deletion by ancestor—a script is deleted by an ancestor script, which (indirectly) includes the deleted script into the frame; 3) deletion by descendant—a script is deleted by a descendant script, which is (indirectly) included by the deleted script into the frame; 4) deletion by other scripts—a script is deleted by a script that has no inclusion dependency with it.

The first three classes are quite close, because the deleting script and the deleted script are dependent on each other. It is likely that a developer separates his code into multiple scripts which successively include each other into the frame and he uses one of them to delete the other(s). In contrast, the last class is quite different, as neither of the two scripts depend on the other. However, it is still possible—although less likely—that both scripts are written by the same developer and are included by the same parent script. As a result, we cannot confidently say that the two scripts in the fourth class are not functionally related.

## 3.2  Script Deletion Techniques

We present the general techniques for deleting a script from a web frame using JavaScript. Since client-side JavaScript can be embedded into a web page through mainly two ways, we discuss separately the corresponding deletion techniques.

*3.2.1  Deleting Host Element.* Most client-side JavaScript code is embedded in the HTML `<script>` elements/tags, either statically by the first-party developer or dynamically by another included script. Such a script can be deleted from the page by deleting its host `<script>` element from the DOM tree. Several DOM APIs can be called in JavaScript to directly remove an element from the DOM tree. They include `Element.remove()`, `Element.replaceWith(node)`, *etc.*

A `<script>` element can also be deleted by removing its parent or ancestor element (*e.g.*, a `<div>` element) from the DOM tree. Furthermore, a script can set the `innerHTML` property of an element (*e.g.*, a script's parent element) to change its contained HTML markup. Similarly, setting the `outerHTML` property of an element (*e.g.*, a `<script>` element or its parent) replaces the element with the DOM nodes parsed from the string operand.

*3.2.2  Deleting Host Attribute.* Client-side JavaScript code can also be embedded within certain attributes of HTML elements as inline scripts. For instance, in `<body onload="console.log('x');">`, the string value of the `onload` attribute of the `<body>` element would be evaluated as JavaScript code when the `load` event of this element is fired. Such

an inline script can be deleted by removing or modifying its host attribute. Note that these scripts may also be deleted by removing the entire host elements (§3.2.1). We list two common types of HTML attributes that can embed JavaScript code below.

- Inline Event Handlers. The HTML standard [36] specifies that event handlers can be exposed as *event handler content attributes*. They are also known as the "on- attributes" or the "inline event handlers", and can be specified for event handlers on HTML elements. JavaScript code contained within these attributes is parsed and executed when the corresponding event handler is invoked.
- URL Attributes. Some HTML element attributes take URLs as values. The browser would navigate to or submit a request to the corresponding URL under certain conditions, *e.g.*, when a user clicks a hyperlink `<a href="URL">...</a>` the browser would navigate to the URL specified in the `href` attribute. The URL can be a *javascript: URL*, that represents a JavaScript script. When the browser executes a *javascript: URL* request, it removes the leading "javascript:" scheme string, and evaluates the rest string in the URL as JavaScript code [36].

However, unlike the code embedded in `<script>` elements, the inline scripts embedded in HTML attributes may not be executed automatically. It would be impractical and very challenging to trigger all such scripts [12, 18]. Thus, we target at the scripts that are automatically executed upon page loading, *e.g.*, the "onload" inline event handlers of HTML elements.

## 3.3  Security Impact

We analyze the security impact of script self-deletion in this subsection. We first classify the self-deletion using the criteria in the prior work [26]. Then we evaluate two state-of-the-art malware detection or analysis tools to investigate current limitations. Readers can refer to the Appendix A and Appendix B for detailed results. Our analysis reveals that JavaScript self-deletion is an anti-debugging technique that tries to prevent, or at least impede, any attempts at (manually) inspecting and debugging the JavaScript code on a website. Since there can be numerous scripts on one page, inspecting the execution of a specific self-deleting script is challenging, especially when it is generated dynamically. Analysts need a heavy work to locate the original code where the self-deleting script is included or generated among thousands lines of code. Unlike static code transformation techniques, in particular obfuscation, self-deletion affects the dynamic code analysis at runtime. Even though the current dynamic malware analysis tools are able to report the malicious activities observed on one website, the security analysts still need to manually locate the real culprits among a huge number of scripts because of the coarse-grained behavior attribution of the current tools.

# 4  DETECTING SCRIPT DELETION

In this section, we present our techniques for detecting the client-side JavaScript script deletion operations. Specifically, we develop a browser-based JavaScript runtime monitoring system—JSRay—for studying client-side script deletion.

## 4.1 Overview

We present the design and implementation of JSRᴀʏ, a browser-based JavaScript runtime monitoring system, to study client-side script deletion. Specifically, JSRᴀʏ incorporates a dynamic script inclusion monitoring mechanism to help track the origin of an included script (§4.2), and a runtime deletion monitoring mechanism to identify the deleting script and the deleted script (§4.4). Additionally, it also dynamically monitors JavaScript's access to sensitive browser APIs for helping to detect potentially suspicious behaviors (§4.5). We decide to take a dynamic runtime monitoring approach in our design and build a prototype of JSRᴀʏ based on Chromium (§4.6). As we had discussed in §2.2, the dynamic features of JavaScript make it difficult to statically analyze the behavior of individual scripts.

We introduce our dynamic script monitoring approach in detail next.

## 4.2 Dynamic Script Inclusion Monitoring

Since a lot of scripts can be dynamically included into a web frame, we need to identify the inclusion relationship between scripts, which we use to determine if a deletion operation is self-deletion or not (§3.1). Prior work [13, 21, 34, 35] describes the inclusion relationships of dynamic elements and analyzes the security problem in inclusion chains. However, their work focuses on the inclusion of first-party or third-party resources loaded on the page, *i.e.*, resources have specific URLs and are included on the page. But it would also be necessary for tracking the origins of the scripts that do not have a source URL, including the dynamically included inline scripts and the dynamically generated scripts via functions like `eval`(). To this end, we build our own inclusion tree by changing the browser to monitor a few important JavaScript and DOM APIs that can be leveraged for script inclusion or script generation.

*4.2.1 Scripts Embedded in Elements or Generated via eval().* First, we identify the scripts that are dynamically included via the HTML `<script>` elements by other scripts, which are their *parent* scripts. Specifically, we need to monitor the script element creation APIs to identify the parent script that is creating a new script element. This can be done by hooking all the DOM APIs that can be used for creating new `<script>` elements, such as `document.createElement("script")` and `document.write("<script>...</script>")`, and checking the JavaScript call stack to find the parent script, as has been demonstrated by the prior work JSIsolate [38].

However, we take a different approach to monitoring script creation in our research because the existing approach is incomplete. Although JSIsolate can track the creation of new HTML `<script>` elements, it cannot monitor the JavaScript code that is dynamically generated through the `eval`() JavaScript function, which is not a DOM API. For the sake of completeness, we instead modify the JavaScript engine's parser to monitor the calls of the JavaScript parsing function. The JavaScript parser must be invoked when any JavaScript code is about to be executed. Therefore, monitoring the JavaScript parsing function would also allow us to identify the parent script when an HTML `<script>` element is being dynamically created. To identify the origin of a dynamically included inline script element, we set its parent script's source URL (if available) as its origin, following the prior work [38].

*4.2.2 Scripts Embedded in Attributes.* Second, we identify the inline scripts that are included via HTML attributes, including the inline event handlers and the *javascript: URLs*. We could have also used the above parser monitoring method to track the inclusion of such scripts because it covers all executed JavaScript code. However, we are unable to identify the parent scripts of the scripts dynamically included via HTML attributes using this method, because such scripts are usually not automatically executed immediately when they are inserted into the DOM tree. In other words, the inclusion of such scripts and their execution are asynchronous.

To track the creation of inline event handlers, we monitor the calls of the browser's internal event handler registration function. This function is called whenever a new event handler is being added in the browser. Similarly, we examine the JavaScript call stack to locate the parent script, if any. However, the event handlers will only be executed when the corresponding events are fired. Therefore, we also monitor the writes to the inline event handler attributes and record the source code for mapping the JavaScript objects to their DOM containers, which we will discuss in §4.3.

To support *javascript: URL* scripts, we also monitor all the writes to the relevant HTML attributes as JSIsolate [38]. We identify the scripts that are setting the corresponding monitored attributes as the parent scripts. Since an attribute could be set by multiple scripts, we identify the parent script from the last write log.

## 4.3 JavaScript and DOM Mapping

We need to map the JavaScript object parsed and executed in the JavaScript world to its original container in the DOM tree, because they are managed and monitored separately and even asynchronously. This would also be needed because the deletion of a script is the removal of its container in the DOM tree instead of its JavaScript object.

Each JavaScript object is labeled with a unique ID in the JavaScript world; we log this script ID to identify the script. We tag the script ID of a script to its container when the script is either being included into the DOM tree or being parsed by the JavaScript parser. To identify the external `<script>` element, we map using the script's source URL. To map the inline scripts, we compare the source code parsed in the JavaScript parser with the source code set in the corresponding element or attribute. Specifically, when an inline script is being included, we would record in our monitoring code its source code obtained from the DOM tree. Since the parsing of a script could be asynchronous with its inclusion into the DOM tree, the mapping is lazily performed when the script is being parsed. For example, the JavaScript object of an (inline) event handler is only created when the corresponding event is fired (for the first time); we therefore perform the mapping in the browser's inner event dispatching function [1].

## 4.4 Script Deletion Monitoring

We monitor the calls of the corresponding DOM APIs to detect the script deletion operations, because the deletion of a script is the removal of its code container in the DOM tree. We had discussed the general techniques that can be used for script deletion in §3.2.

---

[1]https://dom.spec.whatwg.org/#concept-event-listener-inner-invoke

Accordingly, we hook the DOM APIs for deleting the host elements or the host attributes from the DOM tree.

We insert dynamic monitoring code into all the DOM APIs that can be used for element deletion. The monitoring code identifies the target host `<script>` element as the *deleted script*, and locates the *deleting script* that invokes the deletion API from the JavaScript call stack.

To detect the deletion of the host attribute of an inline script, we hook the DOM APIs that can be used for deleting or modifying HTML attributes. We also track if the corresponding inline script has been executed. If not, we do not consider the script as a deleted script. Further, the inline scripts embedded in HTML attributes can also be indirectly deleted by deleting their host attributes' host elements. Therefore, when an element is being deleted from the DOM tree, we additionally check if it contains any embedded inline scripts. The check is also recursively applied to any child elements that are being deleted together.

## 4.5 Sensitive API Access Monitoring

We suspect that some scripts use self-deletion for hiding their suspicious or even malicious behaviors from the users and reducing the chances of being discovered by researchers. Therefore, we also dynamically monitor the operations of JavaScript code. This further enables us to study the connection between the uses of the self-deletion technique and suspicious activities in §6. Our dynamic API monitoring method can overcome the limitation of static methods in analyzing obfuscated code, which makes it difficult to detect the (sensitive) API access patterns.

We focus on two classes of sensitive browser APIs, together with network access APIs in our research.

The first class of sensitive APIs is related to client-side storage. The JavaScript security policy permits all scripts running in the same frame to access these storages associated to the frame. The client-side storage in modern browsers includes Cookies, LocalStorage, SessionStorage, IndexedDB [2] They can be used for storing sensitive user data, such as Cookies, account credentials, *etc.* Therefore, the accesses to the storage APIs are sensitive, yet not necessarily suspicious because the first-party scripts may need the access to provide the necessary functionalities. However, the accesses made by third-party scripts could be suspicious, especially by the self-deleting scripts. Therefore, we monitor all scripts' accesses to those APIs, including the call stack snapshot, script ID and arguments, for further analysis.

The second class of APIs is privacy-sensitive. There exist many browser APIs that can reveal sensitive information about the user and/or the device. Some APIs might even be exploited for side-channel attacks [25, 27] that can leak sensitive user information such as phone image or browsing history. For example, the `Navigator.getBattery()` method provides information about the system's battery. It has been shown that the battery capacity, as well as its level, expose a fingerprintable surface that can be used to track web users in short time intervals [28]. Some sensitive APIs like `MediaDevices.getDisplayMedia` require the user to explicitly grant the permission; we do not monitor the uses of such APIs.

---

[2]WebSQL had been deprecated by the W3C.

Additionally, we monitor APIs that can make requests or send data to other websites, such as `XMLHttpRequest`. Since it is difficult to perform dynamic taint analysis in browsers and the data sent might be encrypted, we currently do not track the data flow from other sensitive APIs. Alternatively, we utilize this as an indicator to demonstrate that the script has a proclivity to access the network when invoking certain sensitive APIs.

## 4.6 Implementation

We implemented a prototype of JSRay based on the Chromium browser version 96.0.4664.45. We modified its rendering engine and the V8 JavaScript engine using about 700 lines of C++ code. The monitored operations are logged in files for offline analysis. We anonymously release the source code [2] for facilitating the research in client-side JavaScript.

## 5 MEASURING SCRIPT DELETION IN THE WILD

In this section, we apply our script-deletion detection system in a large-scale measurement of one-million popular websites to study the prevalence of script deletion behaviors in the real world. Through the measurement, we hope to first answer the following research questions:

- RQ1: how widely the script deletion technique has been adopted in real websites;
- RQ2: whether most scripts are deleted by themselves;
- RQ3: which kind of scripts are deleted.

We will further analyze the script deletion behaviors and their security implications in §6.

## 5.1 Experiment Setup and Dataset

We apply JSRay to detect script deletion on the top one million domains in the Tranco list [22], which is a top domain list that has been commonly used in web security research. To automate the measurement, we drive 20 JSRay instances in parallel using Selenium [3] to visit the main page of each website on the list. In our experiment, we try to load a web page within a 30-second timeout in one attempt and will give up on visiting a website after three failed attempts. If a page can be loaded within 30 seconds, we collect the monitoring data for 10 seconds and clear the user profile. During the visit, we do not generate any user actions on the page to collect the default behaviors of the automatically-executed scripts. The experiment was performed in May 2023.

We were able to collect valid data from 870,734 websites (87.1%) in our measurement. The remaining websites either cannot be reached or did not respond within the timeout. Also, we were unable to reach some websites that were hosted on the same CDNs, which blocked our visits possibly for high volume of requests. Nevertheless, we believe that the 87.1% coverage of the top one-million domains is sufficient for our research.

## 5.2 Prevalence of Script Deletion

We find that script deletion is common in the real world and most deleted scripts are self-deleting scripts. JSRay detected in total

---

[3]https://www.selenium.dev/

**Table 1: Breakdown of the 4th class deletions by privilege group. 1P/3P denotes the first-party/third-party script.**

| Group | #Cases | %Cases | #Websites | %Websites |
|---|---|---|---|---|
| 1P deleting 1P | 1,871,822 | 53.59 | 137,621 | 41.23 |
| 3P deleting 3P | 53,732 | 1.54 | 8,387 | 2.51 |
| 1P deleting 3P | 16,662 | 0.48 | 1,882 | 0.56 |
| 3P deleting 1P | 1,550,328 | 44.39 | 185,898 | 55.69 |

5,089,340 script deletions on 369,525 websites, or 42.44% of the 870K reachable websites. These deletion operations are from 761,240 scripts, and 240,478 unique ones deduplicated by the hash value of the source code. Among the 5,089,340 deletion operations, 1,596,796 (31.38%) were self-deletion, which were found on 226,854 (61.39%) websites. The operations come from 311,402 scripts and 142,557 unique ones. We further categorize script deletion into the four classes defined in §3.1. Among all these deletions, 1,451,167 (28.51%) scripts were deleted by themselves; 142,697 (2.80%) scripts were deleted by ancestor scripts; 2,932 (0.06%) scripts were deleted by descendant scripts; and 3,492,544 (68.62%) scripts were deleted by other irrelevant scripts.

It is a bit surprising that about 69% of the deletions were in the fourth class. We try to understand why those scripts were deleted by scripts without inclusion dependency. We further group the deletions by the privilege (*e.g.*, first-party or third-party) of the deleting script and the deleted script in Table 1. We find that in about half (54.07%) of the fourth-class deletions the deleting script was a first-party script. Since all (third-party) scripts are indirectly included by the first-party developers who should have full control of the web page, it is understandable that a third-party script might be deleted by another first-party script that does not directly include the deleted script. For instance, the first-party developer may use one script to include some other scripts, and use another one to delete the included scripts. Nevertheless, there were still 53,732 (1.54%) cases that a third-party script was deleted by another inclusion-independent third-party script. There were also 1,550,328 (44.39%) cases that a first-party script was even deleted by an inclusion-independent third-party script. By manually analyzing some of the samples, we find that the third-party deleting scripts were usually library functions that provided utilities for managing the DOM tree or the scripts. For example, jQuery provides a lot of library functions for conveniently manipulating the DOM tree. Many of the deletions resulted from the calls of such utility functions by other scripts, which we did not identify as the deleting scripts. This also suggests that many of the fourth-class deletions could have actually been classified as self-deletions if we changed the way to identify the deleting scripts. However, we currently cannot manually confirm that all the 1,550,328 are like these. Additionally, it is challenging to distinguish whether third-party scripts actually delete some scripts or just provide functions for other scripts. We leave it as a future work to distinguish the legitimate library scripts from the real deleting scripts.

Since more than 30% of the deletions are self-deletions and some of the 4th class cases might also be self-deletions, we focus on studying the self-deleting scripts next.

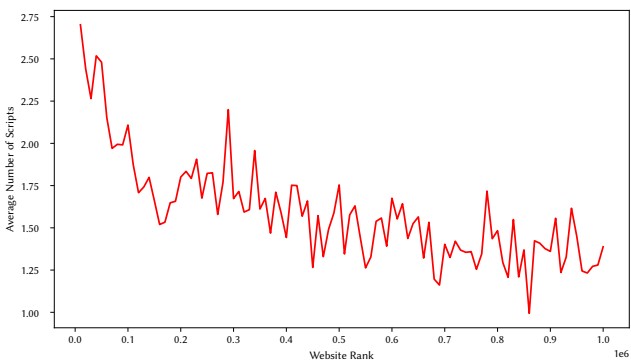

**Figure 1: Average number of self-deleting scripts across websites in different ranking groups.**

## 5.3 Characterization of Self-Deletion

We further characterize the self-deletion behaviors with respect to the distributions of the websites, scripts, and techniques.

*5.3.1 Websites with Self-Deleting Scripts.* We first study the 226,854 websites that included self-deleting scripts. We find that these websites are almost uniformly ranked—we do not observe that some particular ranking group contains more websites than other groups.

However, we do observe that the websites might include quite different numbers of such scripts. We show the average number of included self-deleting scripts of websites in each ranking group in Figure 1. The size of the ranking group is 10,000. In general, there were more self-deleting scripts in more popular websites: the websites in the highest ranked group had on average 2.70 self-deleting scripts, while the ones in the lowest ranked group had only 1.38 such scripts on average. One possible explanation is that the popular websites included more scripts. In other words, the numbers of self-deleting script are positively correlated with the numbers of all included script on the websites. The Pearson's correlation coefficient between the two number series is 0.7007, confirming our hypothesis.

*5.3.2 Self-Deleting Scripts.* Next, we study if most of the self-deletions were related to a small group of scripts, that might be included by many websites. This is likely because we find deletions from 311,402 scripts, while the unique ones are only 142,557. We find that 95.61% of the scripts were found on only one website and the rest 6,262 scripts constituted 57.47% of the self-deletion cases. This explains why the number of self-deletions is much larger than the unique number of the self-deleting scripts.

The top self-deleting script from www.googletagmanager.com was included by 47.28% of the websites. It alone accounted for 475,056 (29.75%) self-deletions in our dataset. This Google Tag Manager script provides service for managing custom tags (scripts) used for advertising, remarketing, analytics, *etc.* It allows developers to insert customized scripts into the page by using an individual `<script>` tag. However, it is commonplace for developers to remove the tag after the injection in order to maintain the structure of the original DOM tree for the sake of compatibility (§6.1). We further discover that the Google Tag Manager script was the only self-deleting script on 73,197 websites.

Table 2: Breakdown of script self-deletions grouped by deletion technique (*Element/Attribute*), script privilege (*1P/3P*) and script inclusion method (*Static/Dynamic*).

| Deletion Group | #Cases | %Cases | #Websites | %Websites |
|---|---|---|---|---|
| Element/Static | 793,281 | 49.68 | 146,771 | 64.70 |
| Element/1P-Static | 237,154 | 14.85 | 26,891 | 11.85 |
| Element/3P-Static | 556,127 | 34.83 | 119,880 | 52.84 |
| Element/Dynamic | 154,769 | 9.69 | 60,845 | 26.82 |
| Element/1P-Dynamic | 71,318 | 4.47 | 21,077 | 9.29 |
| Element/3P-Dynamic | 83,451 | 5.23 | 39,768 | 17.53 |
| Attibute/Static | 642,435 | 40.23 | 79,634 | 35.10 |
| Attribute/1P-Static | 406,976 | 25.49 | 59,863 | 26.39 |
| Attribute/3P-Static | 235,459 | 14.75 | 19,771 | 8.72 |
| Attribute/Dynamic | 6,311 | 0.40 | 1,310 | 0.58 |
| Attribute/1P-Dynamic | 6,207 | 0.39 | 1,289 | 0.57 |
| Attribute/3P-Dynamic | 104 | 0.01 | 21 | 0.01 |

*5.3.3 Self-Deleting Techniques.* We investigate the techniques that were used for script self-deletion by different kinds of scripts and different privileges. Accordingly, we classify the self-deletion cases into eight groups and list the results in Table 2. Note that one website may contain multiple types of cases, therefore the total percentage of websites in all groups is over 100%.

First, we find that the scripts were more frequently deleted by deleting their host elements. They constitute 59.37% of the cases. Second, most websites include self-deleting scripts hosted at third-party servers. Across all technique and inclusion groups, there are more third-party scripts than first-party scripts. Third, most of the deleted scripts were statically included into the websites, regardless of their privilege level and the kind of their host containers. Since dynamic scripts are more difficult to write and debug, it is reasonable that most of the scripts are static ones. However, it is alarming that the existence of these dynamically included and then automatically deleted scripts could hardly be noticed, by not only normal users but also experienced developers. They neither appear in the initial HTML source code returned from the first-party server, nor can be spotted in the rendered DOM tree of the page in the browser developer tool panel. We even discovered cases where a group of scripts were iteratively included through multiple layers and the first script deleted the deeply nested innermost scripts. This makes it very difficult to completely analyze the behaviors of such scripts.

In summary, our results show that self-deletion is quite common on the web. Since it is also a very powerful client-side anti-analysis technique, we demonstrate the strong need of the dynamic JavaScript monitoring systems like JSRay for web security research and client-side JavaScript analysis.

## 6 UNDERSTANDING SCRIPT SELF-DELETION

We have shown that script deletion especially self-deletion was common in real websites. In this section, we aim to study why the self-deletion technique was used by web developers. In particular, we would like to understand whether those self-deleting scripts were connected to suspicious activities. To this end, we first conduct a manual code analysis to figure out deleted contents and explain the reasons for script self-deletion (§6.1). As we discussed in §2.1, the suspicious or malicious scripts we targeted on differ from that of traditional malicious JavaScript detectors. We then leverage our sensitive API monitoring data and existing popular filter lists to

study if self-deleting scripts perform much more suspicious behaviors than other normal scripts §6.2. We finally discuss an interesting case of self-deleting scripts (§6.3).

### 6.1 Deleted Contents and Reasons

We are quite interested in answering what are deleted and why so many scripts employed self-deletion. Because of the limitation of existing malware analysis tools (§3.3), we manually analyzed a small set of 600 unique randomly sampled self-deleting scripts from unique third-parties. Specifically, we read the logged source code of the deleted scripts and the corresponding deleting scripts. We also check how they interacted with the embedding web page and other scripts. It took one of the authors 40 hours to complete the manual analysis. An automated method to detect malicious scripts may facilitate our analysis, but is an orthogonal and challenging research problem. Due to the nature of this new problem and the difficulty of analyzing client-side JavaScript code that could use many different obfuscation techniques, we do not claim that our explanation correctly reflects the real intention of the developers. Rather, we hope that our initial manual investigation could shed light on this emerging problem and facilitate future research.

**Hiding/Cleaning benign code.** First, we discover that the majority (552 or 92%) of the 600 scripts were benign code. Developers can use this technique to clean up temporary code and/or protect their intellectual property. Temporary code refers to first or third-party scripts that are dynamically loaded and executed only once upon creation (but might be created multiple times). It is no longer needed after the execution. Developers include temporary code to provide support for customized contents by invoking the `eval()` function or embedding a `<script>` tag onto the page. However, creating a new `<script>` tag changes the structure of the DOM nodes. It may cause some compatibility issues if other scripts rely on the DOM structure to select the necessary nodes. Therefore, developers tend to initiatively remove the `<script>` tag once inserted. For instance, the jQuery library has a method `html()` that helps developers to insert HTML fragments. However, according to the WHATWG standard [4], script elements do not execute when parsing the fragments. This indicates that methods like `innerHTML` cannot execute any embedded script tags, which is inconvenient in some cases. jQuery extends the method by inspecting the content and leveraging a `DOMEval` function [5] to execute the scripts. The function generates a new `<script>` tag and uses `doc.head.appendChild(script).parentNode.removeChild(script);` to append the script to the `<head>` element and remove it immediately to prevent any side effect upon returning. Therefore, executing the embedded scripts in this manner does not introduce any side effects to the final DOM, such as a new child being appended to the `<head>` element. Please note that the developers do not provide an explanation for why they made this design decision, but we believe that our interpretation is the most convincing explanation. It can also be considered as a common and legitimate modern software development practice to protect one's own software from being copied and plagiarized. By applying self-deletion alone or together with code obfuscation and other anti-analysis techniques, the developers can

---

[4]https://html.spec.whatwg.org/#other-parsing-state-flags
[5]https://github.com/jquery/jquery/blob/main/src/core/DOMEval.js

prevent most users from locating the core code and increase the difficulty of reverse engineering.

**Hiding suspicious activities.** Second, 48 (8%) scripts deleted themselves also for preventing code analysis but performed suspicious or even malicious operations, including accessing and stealing user credentials stored in client-side storage, tracking user activities without user consent, or making profits through immoral ways (*e.g.*, malvertising [23]). Even though some of the suspicious activities could be noticed by the victim users, it is difficult to identify the culprit scripts which were removed from the DOM tree. Like legitimate scripts, these suspicious scripts could also leverage other anti-analysis techniques to further escape the detection of their suspicious behaviors. Although the proportion of such scripts is low in our manual analysis, it is alarming that self-deletion has also been exploited by the web attackers to hide their suspicious/malicious activities.

In summary, we find that most scripts deleted themselves for legitimate purposes like code cleaning or intellectual property protection, yet the self-deletion technique has also been used for hiding suspicious/malicious script behaviors. This motivates us to further study the connection between self-deleting scripts and suspicious operations next.

## 6.2 Self-Deletion and Suspicious Activities

Our manual analysis revealed that some scripts hid their suspicious behaviors by deleting themselves. We leverage the sensitive API monitoring data and a popular filter list to study the behavior difference between self-deleting scripts and the normal scripts. Since Google Tag Manager allows the user to embed custom scripts, these sensitive API accesses are counted from the custom scripts instead of GTM itself. Specifically, we sample 226,854 websites that did not contain script deletion and use all their executed scripts as the normal script dataset, which contains 3,223,953 scripts in total.

We monitored two classes of sensitive APIs, together with network access APIs in our system §4.5. The results are shown in Table 3 in the Appendix. We observe that the normal scripts made on average 2.38 sensitive API accesses, which is much smaller than the average number of accesses—28.12—made by the deleting and deleted scripts across all groups. This demonstrates that the self-deleting scripts in general were more likely to access sensitive data in the browser. By employing the self-deletion technique as an anti-analysis method, their accesses are much more suspicious. The average number of network access APIs made by the self-deleting scripts (0.877) is also 6.1x more than the normal ones (0.143). This suggests that their accesses have a propensity to coincide with network requests, which could potentially result in data leakage.

We then use EasyList (with optional EasyPrivacy supplementary) [6], one of the most popular filter lists used for blocking unwanted web content such as advertisements and trackers, as the ground truth to study whether self-deleting scripts are more likely to be blocked than normal scripts. The result shows that 124,386 (39.94%) self-deleting scripts are among the block list, compared to 229,738 (7.13%) normal scripts. It indicates that self-deleting scripts are much more likely to be blocked by filter lists. In other words,

---

[6]https://easylist.to/, version 202306271740

they are more likely to be related to unwanted web content that many users would like to block.

In summary, we discover that the self-deleting scripts made more accesses to sensitive browser APIs on average than the normal scripts. They are also 5.6x more likely to be included in popular filter lists. Although not all the accesses are necessarily malicious, the uses of script deletion technique increase the difficulty in analyzing their operations. Our research makes an important first step in understanding their behaviors, and lays the necessary foundation for future research on studying the potential malicious activities in such scripts.

## 6.3 Case Study

We discuss one interesting self-deleting case we detected to help further illustrate the use of self-deletion technique in the real world. We suspect that the case was related to malvertising and user data collection without user consent.

In our experiment, we found a group of websites that embedded similar self-deleting scripts. These self-deleting scripts would redirect the visitors of the embedding sites to other landing pages to promote other contents, of which some were malicious, such as malware or online scams. We thought that these websites participated in the same online promotion/advertising network which provided the self-deleting redirection scripts for making profits. We were able to reconstruct all the deleted scripts and hence study their activities with the help of JSRay.

We present one example of such self-deleting scripts in Listing 1 in the Appendix. The original scripts were obfuscated and they leveraged `eval()` to dynamically execute additional code. The script implements a `track()` function to send sensitive user data (*e.g.*, the hostname of the page the user is visiting) it collects to the server of the promotion network—https://t.rainide.com/. It dynamically loads a new script from the server by attaching the sensitive user data as request parameters. It then automatically deletes the newly included script by listening for the `"load"` event of the script (lines 9-14). Finally, the script redirects the user to the product link automatically.

To hide its operations, in addition to self-deletion, this script employed multiple anti-analysis techniques such as dynamic code generation using `eval()` and code obfuscation, including string removal, variables renaming, *etc.* The snippet shown in Listing 1 was actually generated via `eval()`. Consequently, a victim user cannot easily notice those hidden suspicious behaviors.

## 7 CONCLUSION

In this paper, we systematically studied the current client-side JavaScript self-deletion behavior in the real world. We overcame several challenges in studying the dynamic behaviors of JavaScript programs, and developed JSRay, a browser-based JavaScript runtime monitoring system. With JSRay, we conducted a large scale measurement of one million websites. Our experiment results revealed that script self-deletion is quite common in the real world, and that self-deletion has already been employed for hiding suspicious operations in client-side JavaScript. Our work demonstrated the strong need of the dynamic JavaScript monitoring systems like JSRay for web security research.

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

**Table 3: Sensitive API accesses (Average/Average with network access).**

| Group | Total | #Storage | #Privacy |
|---|---|---|---|
| 1P deleting 1P | 5.10/3.46 | 4.60/3.05 | 0.50/0.41 |
| - Deleted script | 2.40/1.62 | 2.17/1.43 | 0.23/0.19 |
| - Deleting script | 2.70/1.84 | 2.43/1.62 | 0.26/0.22 |
| 3P deleting 3P | 47.43/28.41 | 44.33/26.71 | 3.10/1.71 |
| - Deleted script | 23.72/14.21 | 22.17/13.35 | 1.55/0.85 |
| - Deleting script | 23.72/14.21 | 22.17/13.35 | 1.55/0.85 |
| 1P deleting 3P | 0/0 | 0/0 | 0/0 |
| 3P deleting 1P | 3.70/2.44 | 3.34/2.19 | 0.37/0.25 |
| - Deleted script | 0.01/<0.01 | 0.01/<0.01 | <0.01/<0.01 |
| - Deleting script | 3.69/2.44 | 3.33/2.19 | 0.36/0.25 |
| Normal script | 2.38/1.17 | 1.82/0.96 | 0.56/0.21 |
| - First party | 0.94/0.61 | 0.72/0.50 | 0.22/0.11 |
| - Third party | 1.43/0.56 | 1.10/0.46 | 0.33/0.10 |

**Table 4: Systematization of BADTs. The goals are to *Impede* dynamic analysis, subtly *Alter* its results and *Detect* the presence of the analysis. A filled circle means the property fully applies, a half-filled circle means it applies with limitations.**

| Technique | Goal | Effective | Stealthy | Versatile | Resilient |
|---|---|---|---|---|---|
| SELFDELETION | A/I | ◐ | ● | ● | ◐ |
| SHORTCUTx | I | ○ | ○ | ○ | ○ |
| TRIGBREAK | I | ● | ○ | ○ | ◐ |
| CONCLEAR | I | ◐ | ○ | ○ | ○ |
| MODBUILT | A/I | ◐ | ◐ | ● | ◐ |
| WIDTHDIFF | D | ◐ | ● | ● | ○ |
| LOGGET | D | ● | ◐ | ● | ● |

```
1   var track = function (m, query) {
2       var projectId = 601;
3       var url = "https://t.rainide.com/" + projectId + "?metric="
          + m + "&value=1&hostname=" + location.hostname + "&" +
          query,
4           script = document.createElement("script"),
5           id = '_' + Math.random().toString(36).substr(2, 9);
6       script.setAttribute("src", url);
7       script.setAttribute("async", !0);
8       script.setAttribute("id", id);
9       try {
10          script.addEventListener("load", function () {
11              var el = document.getElementById(id);
12              el && (el.remove ? el.remove() : el.parentNode &&
          el.parentNode.removeChild(el))
13          })
14      } catch (ex) {}
15      document.head.appendChild(script);
16  };
17  /* some supporting functions */
18  track('pingMe', /*sensitive data*/);
19  window.addEventListener("load", function (event) {
20      /* get metrics and statistics */
21      track('pingMePushka', /*sensitive data*/);
22      track('superTimings', jsonToQueryString(window.performance.
          timing.toJSON()));
23  });
24  /* redirect to promotion link */
```

**Listing 1: Part of self-deleting scripts from https://goldclassmusic.com (recovered).**

## A  BADT CLASSIFICATION

We examine self-deletion and compare it with 6 known BADTs systemized in the prior work [26]. The results are shown in Table 4. Self-deletion is effective for hiding scripts that their DOM nodes are removed from the DOM tree and cannot be easily recovered. The browser does not prompt any notice to users or throw any warnings when the scripts are removed from the DOM tree. Furthermore, as we have shown in §3.2, there are plenty of variants of this technique. There is no good mechanism to prevent the deletion operation or recover the deleted scripts. The trace left by a self-deleting script can only be observed in the incoming network responses, which may contain many different scripts and other web contents. Given that the self-deleting scripts can be dynamically generated via methods like `eval()`, this further makes the identification and analysis of the self-deleting script quite challenging.

## B  STATE-OF-THE-ART MALWARE DETECTION OR ANALYSIS TOOLS

We use two widely used and publicly available tools to study the self-deletion technique: VirusTotal [7] and HybridAnalysis [8]. VirusTotal is an online solution which aggregates the scanning capabilities provided by 89 anti-virus tools, scanning engines, and datasets. It has been commonly used in prior works [13, 16, 17]. HybridAnalysis is another online sandbox widely used by researchers and the industry to conduct malware analysis research and investigation [6]. It loads a target URL in the browser process running in the sandbox and monitors the process's suspicious behaviors such as file system accesses.

We manually selected and analyzed 20 malicious examples from Hynek Petrk JavaScript malware collection [29]. We embedded the malicious code in inline `<script>` tags of the HTML code. All of the samples were identified and marked as *Malicious* by both detectors, serving as the baseline for our analysis. After experimenting with different script inclusion and deletion methods, we found that the state-of-the-art malware analysis tools have the following limitations.

- **Ineffective static analysis technique.** VirusTotal collects results from many static analysis tools, which rely on traditional blacklist or code fingerprinting techniques. They can be easily bypassed by changing the script source URL or applying code obfuscation, making them ineffective in detecting malicious scripts. This indicates that the vendors used by VirusTotal rely solely on the static blacklist technique to identify malicious URLs. In contrast, HybridAnalysis was able to successfully identify the malicious code because it actually fetched and analyzed the script source code.

- **Coarse-grained malicious behavior attribution.** The dynamic malware analysis tool HybridAnalysis employs a traditional process monitor mechanism to detect and report malicious behaviors for the entire website or file. However, we found that it was unable to attribute the malicious behaviors to specific scripts. Such coarse-grained attribution would be undesirable as security analysts need to perform additional heavy manual analysis to identify the real culprits, which are mixed with numerous JavaScript code and other web contents on the page. Script self deletion would significantly complicate such expensive manual analysis for locating the malicious scripts that delete themselves from the DOM tree.

---

[7]https://www.virustotal.com/
[8]https://www.hybrid-analysis.com/

- **Limited scope of monitored behaviors.** Behaviors reported by HybridAnalysis are limited to traditional malicious ones such as reading or writing the file system, and changing system registry records. These malicious operations can be hardly performed in modern browsers that implement various security mechanisms. The more relevant malicious behaviors, such as personal data exfiltration and leakage, are not covered by it.

## C  RELATED WORK

**Anti-Analysis Techniques.**  Malware developers embed multiple anti-analysis techniques in their code to retard the analysis processes by analysts and sandboxes [19]. Malware with self-deletion [7] is an application of anti-analysis technique. Advanced persistent threat (APT) groups [1] are known to conduct targeted campaigns and then self-delete the deployed malware. It involves Anchor [5], which is a backdoor used selectively on high-profile targets that can self-delete its dropper. The security community has also discovered the self-deleting Trojan malware [14]. Some suspicious self-deleting scripts will incorporate obfuscation technique, which is an anti-analysis method [9, 32]. Marius *et al.* [26] introduces 9 anti-debugging techniques and discusses their advantages and drawbacks. In this work, we analyzed the suspicious behavior of self-deleting JavaScript, which is also an anti-analysis technique.

**JavaScript Analysis.**  Many researchers have contributed to JavaScript analysis. Richards *et al.* [30] focused on the security risks by using the `eval()` function. Lauinger *et al.* [21] introduced causality trees to describe the inclusion relationships. Other researchers improved techniques to further study the inclusion relations and explore the maliciousness in inclusion chains [13, 34, 35]. But all these works fail to track the origins of all kinds of scripts such as those which don't have a source URL or are dynamically generated. In [18] and [12], the authors study how to cover and forced execute most JavaScript code for script analysis. Zhang *et al.* [38] proposed JSIsolate, which provides an isolated JavaScript execution environment and studies the dependency relationship between scripts. Based on the JSIsolate technique, we further carried out deeper functional development to study script deletion behaviors. The main ideas of these works are orthogonal to our focus on self-deleting JavaScript.