# OpenReview forum: "Detecting and Understanding Self-Deleting JavaScript Code"
_ACM.org/TheWebConf/2024/Conference — TheWebConf24 Oral_

### Official Review · Reviewer_5iQr · 2023-10-31

**Novelty:** 5
**Technical Quality:** 5

**Review:**

**Strengths**
+ interesting findings about self-deleting scripts on the Web
+ technique to identify self-deleting scripts

**Weaknesses**
- experimental evidence does not support the correlation between self-deletion and malicious intent on the Web
- data collection failed for substantial fraction of sites
- clarification of ethical compliance for non-intrusive data collection
- capturing dynamically generated JS code is not a new contribution
- open-source tooling unclear


Thank you for submitting your work to WWW'24. The paper was an interesting read. In fact, studying self-deleting scripts on the Web is crucial for understanding evolving security threats and protecting against them.

The main strength of this paper lies in the findings about self-deleting scripts on the Web, such as the prevalence of script deletion, the characterization of first-party and third-party scripts involved (important for understanding possible defense mechanisms), and the different types of self-deleting techniques used in the wild. The tooling to identify such self-deleting behaviours is also an added plus.

Another strength of this paper is the surprising findings presented in Section 6.2. The paper effectively establishes a correlation between self-deletion and access to sensitive browser APIs. Moreover, it highlights that self-deleted scripts are more prone to being blocked by ad-blocking lists like EasyList.

However, I also have a few noteworthy concerns:

- First, the results of data collection in Section 5.1 suggests that requests to about 130K domains were blocked, which, according to the paper, were due to the high volume of requests. This brings up two observations: i) what ethical considerations have you taken into account for non-intrusive data collection? ii) the failed sites are a substantial fraction of the dataset (more than one out of every ten), which can introduce potential bias into the derived conclusions.

- An additional concern related to data collection pertains to the methodology employed, in that the study exclusively acquires and analyzes a single webpage from each website (the landing page), raising questions about the dataset's representativeness and comprehensiveness.

- Then, there is also the question of whether self-deletion could indicate maliciousness on the Web. The paper presents weak evidence for a direct correlation between self-deleting scripts and malicious intent. Specifically, the authors reviewed 600 self-deleting scripts in Section 6.1, and found that only 8% of them appeared to engage in suspicious activities. This suggests that self-deleting scripts are more likely to signal benign functionality rather than malicious intent in the context of Web. Furthermore, the quality and scale of this part of the study is also constrained by the limits of manual analysis, i.e., 600 scripts only, hard to reproduce and error-prone.

- In Section 4.2.1, the authors mention that capturing dynamically generated code via eval() is a challenge and create a modified version of a browser engine. However, existing works (e.g., [1, 2]) use the 'Debugger.scriptParsed' event of the Chrome CDP [3] to capture all parsed scripts. This includes all scripts (including those that are dynamically loaded or dynamically generated via string evaluations). Therefore, this is not a new contribution. If the sole challenge addressed by RQ1 in Section 2.2 is the collection of JavaScript source code, it has already been accomplished in previous research.

- Another issue that lies in the intersection of presentation and technical correctness is that lack of clarity regarding what the paper contributes for each of the challenges enumerated in Section 2.2. For example, it is lost on me how the proposed system, JSRay, can handle obfuscated code and in what aspects it differs from existing dynamic analysis and runtime monitoring techniques.

- Finally, it is unclear if JSRay will be open-source to benefit the community.

Overall, I believe the authors have raised interesting points, and the paper holds the potential for acceptance provided that the authors clarify their compliance with ethical standards for non-intrusive data collection.

References
- [1] https://dl.acm.org/doi/10.1145/3372297.3417267
- [2] https://ieeexplore.ieee.org/document/10179403
- [3] https://chromedevtools.github.io/devtools-protocol/tot/Debugger/#event-scriptParsed


## Update After Rebuttal

Thank you for answering my questions. The rebuttal addresses (most of) my concerns. After reading your answers, I have a few recommendations to improve the manuscript:

- Consider incorporating clarifications about CDP methods like Debugger.scriptParsed into the paper. In Section 2.2, the "Dynamic Code" paragraph seem to discuss a challenge addressed in prior work using Debugger.scriptParsed. It focuses on covering dynamically loaded or generated JavaScript code, highlighting the limitation of network request-based approaches due to the absence of a source URL for inline scripts. However, what the paper actually contributes is addressing the challenge of identifying script tag containers (i.e., DOM nodes) on the top of that,

- Consider including a discussion covering ethical considerations (e.g., for data collection) and the limitations of your work (e.g., coverage of crawling, failed requests, etc).

- Please make it clear in your paper that your tool will be open source.

All in all, I think this is an excellent piece of work, and I am happy to recommend it for acceptance.

**Questions:**

- In Section 5.1, what ethical considerations have you taken into account for non-intrusive data collection?
- Will your tool be open source and publicly accessible to benefit our community?
- How does JSRay differ from existing dynamic analysis / runtime monitoring techniques, particularly for handling obfuscated code, as motivated in Section 2.2

**Ethics Review Description:**

Section 5.1 suggests that requests to about 130K domains were blocked, which, according to the paper, were due to the high volume of requests. This introduces the question as to whether the authors employed intrusive data collection and crawling methods?

**Reviewer Confidence:**

4: The reviewer is certain that the evaluation is correct and very familiar with the relevant literature

**Scope:**

4: The work is relevant to the Web and to the track, and is of broad interest to the community

---

### Official Review · Reviewer_7A9o · 2023-11-17

**Novelty:** 6
**Technical Quality:** 6

**Review:**

This paper is undoubtedly interesting and is perhaps the best paper in my pile of papers. The authors propose modifications to the V8 JS engine (+700 lines of code). The authors add hooks to JS code handlers that are responsible for inserting and deleting scripts. They clearly show that merely monitoring the script tags are not enough. There is a need to go under the hood and add annotations. As compared to merely script-based interventions, actually modified the V8 code and then integrating it in Chromium is clearly a very solid contribution.

There are some things that stop the paper from becoming stellar.

1. The scope is somewhat limited because we are not looking at script modification.
2. A lot of subtle bugs are being missed where instead of deleting a script, the environment is changed such that certain parts of the loaded scripts are active and the rest are inactive. This is conceptually similar to deletion but they go undetected in this system.
3. The security of the logging mechanism should be discussed.

**Questions:**

1. How is script modification and changing the control flow in a script by modifying the environment tackled?
2. What is the security of the logging mechanism?

**Ethics Review Description:**

I think it is fine.

**Reviewer Confidence:**

3: The reviewer is confident but not certain that the evaluation is correct

**Scope:**

3: The work is somewhat relevant to the Web and to the track, and is of narrow interest to a sub-community

---

### Official Review · Reviewer_kBNH · 2023-11-18

**Novelty:** 4
**Technical Quality:** 3

**Review:**

This work presents a large-scale analysis of the behaviors of self-deleting Javascript code in the top 1 million pages. The authors present a novel browser-based solution that collects inclusion trees, and more importantly, records deletion behaviors of running JS code. Analyzing the behavior of in-the-wild Javascript with respect to making the web a safer place for consumers remains an important focus of the attention of the security community.

The authors present an evaluation that ~8% of sampled self-deleting scripts perform some kind of malicious activity, which sounds concerning. But the evaluation conflates legitimate issues with questionable security relevant issues. I.e., tracking user activities is a legitimate practices employed by website owners to monetize their business. Similarly, relying on tracking or ad lists to correlate "undesired" behavior does not have immediate security impact. The evaluation of the suspicious activity requires a more fine grained evaluation to allow for proper evaluation of security sensitive vs "undesired" behaviors.

Overall, I feel like the tangible security issues presented by the paper are conflated with non-security issues and that the work might be better suited for a non-security focussed track in its current form.

The paper claims that it is is the first one to investigate the inclusion trees spanned outside of those observable via network requests, which was already done in [34]. Similarly, the authors are distinguishing between 1st and 3rd parties without considering 1st party CDNs etc, similar to the concept of extended same party of [34].

Further analyses that might help shed more light onto the landscape of self-deleting scripts:
- Where do most of the scripts performing the deletion come from? E.g., are those mostly libraries/integration with ad/tracking vendors? E.g., what about a malvertising script distributed via a legitimate ad vendor. We would count the ad as trying to "hide" whereas the distribution mechanism actually performs the deletion as "good" practice to not pollute the DOM.
- What is the exact distribution among the 8% of scripts that are hiding their malicious behavior?
- Does self-deletion happen predominantly together with obfuscation? I would assume that obfuscation is the stronger "hiding" technique?

Pros:
- Jsray is a nice system to understand script inclusion and deletion behavior
- Anti-analysis techniques have the potential to be very security sensitive

Cons:
- security impact is shallow, and needs more detailed/extensive evaluation to shine
- main takeaways fit better into a measurement focussed track than a security track
- techniques are based on an already outdated chrome

Nitpicks:
- use they/them instead of other pronouns for, e.g., the attacker

**Questions:**

- Where are the deletions predominantly happening, e.g., if we see them happening in the head that is part of a different workflow than, e.g., somewhere in the middle of the document?
- Are self-deletion practices correlated with single page applications?
- Why would we employ self-deletion over obfuscation, i.e., looking at an obfuscated piece of JS code feels a lot harder to analyze than a self-deleting one?
- Is self-deletion predominantly a library feature? E.g., beside jquery I would assume that rocket loader and similar technology contributes to the majority of cases.
- If the case study the script performs a redirect anyway, is there a need to perform the self-deletion? E.g., if it is a network script we can easily investigate in the sources devtools tab, or if we pause the script before the navigation we could investigate the DOM for an inline script, or add breakpoints on deletions from the DOM.

**Reviewer Confidence:**

4: The reviewer is certain that the evaluation is correct and very familiar with the relevant literature

**Scope:**

3: The work is somewhat relevant to the Web and to the track, and is of narrow interest to a sub-community

---

### Official Review · Reviewer_CxzJ · 2023-11-22

**Novelty:** 4
**Technical Quality:** 4

**Review:**

Pros:
- Always interesting to see a deep dive into a anti-analysis technique.
- The tool created by the work shows a useful technique for browser instrumentation. The tool and technical measurement approach seems very useful in general, potentially even for other measurements.

Cons:
- It is not clear that the vast majority of website authors are intentionally self-deleting code. The claims made are stronger than the data seems to indicate.
- Unclear what the impact of these findings is, since the self-deleting code was found to be a weaker signal for security-relevant things.

The claims that the work makes in multiple places seems to be to be stronger than the data indicates. For example, in Section 6.2, accessing the listed APIs, network access, and presence on the EasyPrivacy ad blocking list are all things that seem more related to privacy and ad blocking preferences not security impactful concerns. The article assumes and implies that malicious code will score higher on these metrics than benign code, whereas in practice I am not sure if this is the case. The work states that JavaScript self-deletion is an anti-debugging technique, when the results show seem to imply that many website authors may not even know that are creating self-deleting code when they are jQuery or Google tags. In section 6.1, the authors assume that the website authors are intending to use self-deleting code, when the paper doesn't provide any evidence that the majority of these cases are intentionally. I could imagine that website authors just use the library or tool that helps them achieve their goals and don't know or care about the details of how jQuery or the google tags library are implemented.

I would like to see more details provided around the manual classification of benign code and suspicious code in section 6.1. Manual classification into benign and suspicious is very hard to do reliably even for an experienced researcher. Many times, manual researchers just lack the context of what the site's intentions and reasoning are. It would be good to provide explicit definitions for what is suspicious or malicious and what evidence is used to reach that conclusion. Furthermore, statistics for inter-rater reliability rating would be useful here, since this is an area where personal judgment can influence the results. Finally, I'm surprised that the manual classification didn't have any borderline or unknown cases, since, in my experience, there are many situations where it is impossible to know really why a piece of data is being used and many possible suspicious or non-suspicious reasons for it being collected. The manual classification experiment is the only experiment that draws a direct link between this behavior and malicious or suspicious code, so it is important for these findings to be more rigorous.

Finally, I would like more clarification in the work about what the impact of these findings are. It would help if the authors could clarify how the tool or the knowledge gained from the study should impact real systems. I have a vague feeling that the findings are potentially security relevant, but nothing explicit about how the findings of this study can make the state of things better.

**Questions:**

Questions:
- What is the impact of the findings?
- How reliable is the link between self-deleting JavaScript and security?
- How reliable is the manual classification of suspicious and benign JavaScript?

**Reviewer Confidence:**

3: The reviewer is confident but not certain that the evaluation is correct

**Scope:**

4: The work is relevant to the Web and to the track, and is of broad interest to the community

---

### Official Review · Reviewer_pNLR · 2023-11-22

**Novelty:** 6
**Technical Quality:** 5

**Review:**

Pros:

- Well-executed measurement of an interesting phenomenon on the Web. I wasn't even aware that this is done in the wild. Thanks for describing, analyzing and measuring this.

- I also very much appreciate that the authors open-sourced an artifact for this study, such that the reader can check how certain parts are implemented and that future work can benefit from the implementation, thanks.

- The various examples and the case study made it easy to understand the concept and the results of the paper.

Cons:

- My biggest concern here is that although there is a subsection about the potential security impact and an assessment of sensitive API accesses, I don't see much Web security relation in this measurement work. Thus, I would recommend moving this to one of the measurement tracks, e.g. search track or submit it to conferences that target Web measurements e.g. IMC.

- I was wondering about the exact reasons for those self-deleting scripts in the non-malicious use case. For example, in the case of protection of intellectual property, it is weird as someone interested in the code could just directly request it from the source or collect eval invocations via e.g. the Trusted Types API. I understand that a survey study is out of scope here, but asking the Web operators about the reason to get an initial idea for the reason would be a nice addition to the paper.

- It should be made clear why the authors decided to alter chromium instead of using MutationObserver and Trusted Types together with stack traces of the invocations to implement that fully with client-side code instead of altering the execution engine. I'm sure there are reasons for both ways, but the paper would benefit from a more detailed explanation for this choice as it might point out why and in which cases it works better.

- I was very surprised about the high number of sites that have deleting scripts in their application. Although I don't question the correctness of the assessment, it would be nice to have a detailed analysis of the root cause here, e.g. extend the library analysis to see how often libraries are causing that and why they do so. This has already been hinted at with the jQuery case, but I think the paper would benefit from a more detailed analysis here.

- A Minor thing: The eval API seems to be used here as a general representative of string-to-code conversion functions. Also, setInterval or setTimeout can be used to do that, given that the analysis is based on JS parser invocations those cases are not missed, but it should be mentioned somewhere that those are also covered by your tool.

**Questions:**

- Did you contact the developer or operator of the deleting scripts to assess the reasoning behind the deletion, especially for the benign use cases?

- Why exactly did you choose to alter chromium instead of using client-side APIs that cover all cases per default?

**Ethics Review Description:**

No issues

**Reviewer Confidence:**

3: The reviewer is confident but not certain that the evaluation is correct

**Scope:**

2: The connection to the Web is incidental, e.g., use of Web data or API

---

### Decision · Program_Chairs · 2024-01-22

**Decision:**

Accept (Oral)

**Comment:**

## Summary
 This paper conducts a comprehensive study of self-deleting JavaScript behavior on the web, introducing a novel tool, JSRay, to monitor and analyze this phenomenon. The study covers one million popular websites, uncovering the prevalence of script self-deletion, often used for legitimate purposes but also found in conjunction with other anti-analysis techniques in suspicious operations. While self-deletion is identified as a method to evade detection, the study also reveals its use for benign reasons, adding complexity to the analysis of JavaScript security.

 ## Evaluation
 **Strengths:**
 1. **Innovative Research:** The study tackles an underexplored area in JavaScript analysis, focusing on the emerging trend of self-deleting scripts.
 2. **Comprehensive Tool:** JSRay, the browser-based tool developed for this study, effectively captures and analyzes script deletion behavior.
 3. **Significant Findings:** The research reveals that self-deleting scripts are prevalent and used for various reasons, including legitimate ones, challenging traditional assumptions about script deletion in web security.

 **Weaknesses:**
 1. **Scope of Security Relevance:** The paper conflates security issues with non-security issues, lacking a detailed evaluation of truly security-sensitive behaviors.
 2. **Data Collection Limitations:** The study's data collection method failed for a significant fraction of sites, potentially introducing bias.
 3. **Lack of Clarity in Contributions:** The paper does not clearly differentiate its contributions from existing dynamic analysis techniques, particularly for handling obfuscated code.

 ## Suggestions for Improvement
 1. **Distinguish Security Relevance:** Provide a more detailed analysis to differentiate between security-sensitive and non-security issues related to self-deleting scripts.
 2. **Address Data Collection Gaps:** Clarify the impact of the failed data collection on the overall findings and explore ways to minimize biases.
 3. **Clarify Technical Contributions:** Elaborate on how JSRay's approach to monitoring self-deleting scripts differs from and improves upon existing methods.

 ## Overall Impression
 The paper presents valuable insights into the complex nature of self-deleting JavaScript scripts on the web, highlighting both legitimate and suspicious uses. While the study offers significant contributions to understanding JavaScript behavior, it requires refinement in distinguishing between security and non-security aspects and clarifying the novelty of its technical approach.

 Furthermore, given that the connection to web security isn't robustly established, the Program Committee suggests that this paper would be more appropriately situated in the Web Mining and Content Analysis track, rather than the Security and Privacy track. This recommendation is based on the paper's stronger alignment with content analysis and web data mining, rather than direct implications for web security.

 ---